# 17β-Estradiol-Induced Conformational Changes of Human Microsomal Triglyceride Transfer Protein: A Computational Molecular Modelling Study

**DOI:** 10.3390/cells10071566

**Published:** 2021-06-22

**Authors:** Yong-Xiao Yang, Peng Li, Pan Wang, Bao-Ting Zhu

**Affiliations:** 1Shenzhen Key Laboratory of Steroid Drug Discovery and Development, School of Life and Health Sciences, The Chinese University of Hong Kong, Shenzhen 518172, China; yangyongxiao@cuhk.edu.cn (Y.-X.Y.); lipeng@cuhk.edu.cn (P.L.); wangpan@cuhk.edu.cn (P.W.); 2Shenzhen Bay Laboratory, Shenzhen 518055, China

**Keywords:** microsomal triglyceride transfer protein, protein disulfide isomerase, 17β-estradiol, stability, molecular modelling analysis

## Abstract

Human microsomal triglyceride transfer protein (hMTP) plays an essential role in the assembly of apoB-containing lipoproteins, and has become an important drug target for the treatment of several disease states, such as abetalipoproteinemia, fat malabsorption and familial hypercholesterolemia. hMTP is a heterodimer composed of a larger hMTPα subunit and a smaller hMTPβ subunit (namely, protein disulfide isomerase, hPDI). hPDI can interact with 17β-estradiol (E_2_), an endogenous female sex hormone. It has been reported that E_2_ can significantly reduce the blood levels of low-density lipoprotein, cholesterol and triglyceride, and modulate liver lipid metabolism in vivo. However, some of the estrogen’s actions on lipid metabolism are not associated with estrogen receptors (ER), and the exact mechanism underlying estrogen’s ER-independent lipid-modulating action is still not clear at present. In this study, the potential influence of E_2_ on the stability of the hMTP complex is investigated by jointly using multiple molecular dynamics analyses based on available experimental structures. The molecular dynamics analyses indicate that the hMTP complex in the presence of E_2_ has reduced interface contacts and surface areas. A steered molecular dynamics analysis shows that the forces required to separate the two subunits (namely, hPDI and hMTPα subunit) of the hMTP complex in the absence of E_2_ are significantly higher than the forces required to separate the complex in which its hPDI is already bound with E_2_. E_2_ makes the interface between hMTPα and hPDI subunits more flexible and less stable. The results of this study suggest that E_2_-induced conformational changes of the hMTP complex might be a novel mechanism partly accounting for the ER-independent lipid-modulating effect of E_2_.

## 1. Introduction

Human microsomal triglyceride transfer protein (hMTP) is a member of the large lipid transfer protein (LLTP) superfamily which facilitates the transport of lipid molecules between membranes [1,2]. hMTP can transfer triglycerides, cholesteryl esters and phospholipids from endoplasmic reticulum (ER) to primordial apoB-containing particles for the formation of complete apoB-containing lipoproteins, a critical step involved in the assembly of the very-low density lipoproteins (VLDL) and chylomicrons produced in the liver and intestine [3,4,5,6]. During the assembly of lipoproteins, hMTP can deliver triglycerides to nascent apoB molecules through direct binding interactions with apoB [3]. Dysfunction of hMTP could result in several disease states, such as abetalipoproteinemia (ABL), familial hypercholesterolemia, and fat malabsorption [7,8,9]. Because of the unique role of MTP in lipid metabolism, it has become an important novel drug target for lowering blood lipid levels and for treating disorders characterized by the elevated production of apoB-containing lipoproteins, including atherosclerosis, metabolic syndrome, familial hypercholesterolemia and hypertriglyceridemia [8,10,11].

Experimental studies have shown that hMTP is a heterodimer consisting of two subunits: a larger hMTPα subunit and a smaller hMTPβ subunit [6,12,13]. The smaller hMTPβ subunit is the protein disulfide isomerase (hPDI), a member of the thioredoxin superfamily [13,14]. hPDI can catalyze the oxidation and isomerization of disulfide bonds to facilitate nascent protein folding [15,16,17,18]. Studies have also shown that hPDI can bind several small molecules, such as endogenous hormones and environmental compounds [17,18]. 17β-Estradiol (E_2_) is an endogenous female sex hormone that can inhibit the catalytic activity of hPDI [19,20]. It is of note that it has been reported by us and others that E_2_ can significantly reduce blood levels of LDL, cholesterol and triglyceride levels and modulate liver lipid metabolism in vivo, which may partially account for the gender differences in lipid homeostasis [21,22,23,24]. Interestingly, it is apparent that some of the lipid-modulating effects of E_2_ are not mediated by estrogen receptors (ERs) [24]. The exact mechanism by which E_2_ alters lipid metabolism in an ER-independent manner is still not fully understood at present.

Amino acid sequences of human hMTPα subunit and hPDI (hMTPβ subunit) were determined by Sharp et al. [25] and Cheng et al. [26], respectively. The sequence of hMTPα subunit contains 894 amino acid residues, and its signal peptide includes 18 residues [25]. There are 508 amino acid residues in the sequence of hPDI, with 17 residues as the signal peptide [26]. Recently, Biterova et al. reported the crystal structure of hMTP complex consisting of hMTPα subunit and hPDI [27]. The experimental structure of hPDI–E_2_ complex is not available at present. In 2011, our laboratory characterized the E_2_-binding site of hPDI by jointly employing biochemical testing and molecular modelling analysis [28].

In our earlier study, we found that during the binding interaction between E_2_ and hPDI, there is a critical hydrogen bond formed between hPDI HIS256 and E_2_ [28]. Interestingly, in the crystal structure of the hMTP complex reported more recently, it was observed that there is also a hydrogen bond formed between hPDI HIS256 and hMTPα TYR605 [27]. Since hPDI HIS256 can form a hydrogen bond with both E_2_ and hMTPα TYR605, it is speculated that the binding interaction of E_2_ with hPDI might interfere with the binding interaction of hMTPα with hPDI. This possibility is explored in the present study by using molecular modelling analyses that aim to investigate the potential influence of E_2_ on the stability of the hMTP complex, based on the crystal structure of hMTP [27] and the E_2_-binding site structure of hPDI characterized earlier in our laboratory [28]. The modulating effect of E_2_ on the hPDI-associated hMTP complex might help suggest a potential mechanism for its ER-independent lipid-modulating effect.

## 2. Materials and Methods

### 2.1. Construction of the Full-Length hMTP Complex Structure

The crystal structure of hMTP complex presents an extended cradle-like conformation composed of hMTPα subunit and hPDI [27] (Appendix A). The hMTPα subunit contains three domains: β-barrel domain (residues 19-297), α-helical domain (residues 298-603) and lipid-binding domain (residues 604-894) (Appendix A). hPDI contains four major domains: *a* (residues 26-133), *b* (residues 137-232), *b*’ (residues 235-349) and *a*’ (residues 369-479) domains (Appendix A). Two catalytic CXXC motifs (CYS53-GLY54- HIS55-CYS56 and CYS397-GLY398-HIS399-CYS400) are located in the *a* and *a*’ domains of hPDI, respectively [27,29] (Appendix A).

In the experimental structure of the hMTP complex, there are 13 and 30 missing residues in the mature hMTPα subunit and hPDI, respectively [27]. The missing residues (i.e., residues 19, 718-720 and 886-894 of hMTPα subunit; residues 18 and 480-508 of hPDI) were added back using the SWISS-MODEL (https://swissmodel.expasy.org/), a web-based tool for homology modelling [30]. The amino acid sequences of hMTPα subunit and hPDI were downloaded from UniProt [31]. The crystal structure of hMTP complex (PDB code: 6I7S) [27] was downloaded from the Protein Data Bank [32], which was used as a template for homology modelling.

### 2.2. Construction of hMTP–E_2_ Complex Structure

#### 2.2.1. Protein-Ligand Docking

The hPDI–E_2_ complex was predicted based on the known E_2_-binding site in hPDI using the protein-ligand docking method. The procedures are described as follows:

##### Preparation of Protein Structure

hPDI was processed with the Protein Preparation Wizard in the Schrödinger Suite (Maestro 11.9, 2019; Schrödinger LLC, New York, NY, USA). Hydrogen atoms were added and then adjusted for bond orders. The protonation and tautomeric states for residues were adjusted according to a theoretical pH at 7.0. Missing residues and loop segments close to the active site were added back using the Prime (Prime 2.1, 2019; Schrödinger LLC). Water molecules were deleted. Proteins were subjected to geometry optimization using the OPLS3e force field [33].

##### Docking

Schrodinger Glide software is a widely used docking program in drug discovery [34,35]. The commonly used docking method Glide-XP (extra precision) was used in this study, with default docking parameters. Glide-XP docking uses hierarchical filters to find the best ligand binding poses in the defined grid space for a given target protein. The filters include the positional, conformational and orientational sampling of the ligand [36,37,38]. Afterwards, the lowest energy poses were subjected to the Monte Carlo (MC) procedure that samples the nearby torsional minima. The best poses for the given ligand were determined by the GlideScore [39], including terms for buried polar groups and steric clashes. The docking grids for protein structures were generated using Maestro. The grid box was centered at the ligand position in the crystal structures, and the grid box dimensions were set at 20 × 20 × 20 Å^3^.

#### 2.2.2. Structural Alignment

hMTP–E_2_ complex was constructed based on hPDI–E_2_ and hMTP complexes using VMD [40]. The hMTPα subunit in hMTP complex was placed in a similar position as the hPDI–E_2_ complex by superimposing the structures of hPDI in the two complexes. The hMTP–E_2_ complex was further optimized using energy minimization.

### 2.3. Molecular Dynamics (MD) Simulation

In order to investigate the influence of E_2_ on hMTP complex, molecular dynamics (MD) simulations were conducted to characterize the interface stabilities in hMTP and hMTP–E_2_ complexes. The procedure is described as follows:

#### 2.3.1. System Preprocessing

In hMTP and hMTP–E_2_ complexes, backbone atoms of the protein and all the atoms of E_2_ were extracted to avoid local unreasonable structures. The sidechain atoms were added to the backbone using *CHARMM-GUI* (http://www.charmm-gui.org) [41]. This program was also used to generate the topology files of the two systems. The force field parameters of E_2_ were calculated using an antechamber [42]. The *CHARMM36m* force field [43] was adopted for protein. The complexes were embedded into rectangular water boxes extending the solvent 15 Å in x, y, z directions, and the TIP3P water model [44] was used. K^+^ ions with ion parameters approximated by Roux et al. [45] were added to neutralize the charge of the whole systems.

Based on an earlier study [27], a hydrogen bond between hPDI HIS256 and E_2_ is critical for hPDI–E_2_ binding. In the crystal structure of hMTP complex, there is also a hydrogen bond formed between hPDI HIS256 and hMTPα TYR605 [27]. Therefore, to investigate the potential roles of these hydrogen bonds, the following procedures were performed using NAMD [46] in the two systems with two different situations: the hMTP complex, with or without a hydrogen bond constraint between hPDI HIS256 and hMTPα TYR605; the hMTP–E_2_ complex, with or without a hydrogen bond constraint between hPDI HIS256 and E_2_.

#### 2.3.2. Energy Minimization, Equilibrium Simulation and Production Simulation

Firstly, structures of the systems were energy-minimized using the steepest descent algorithm for 10,000 steps. Then, the systems were equilibrated in NVT ensemble for 1 ns, which is called equilibrium simulation. The time step was set to 2 fs. The temperature was maintained at 300 K using Langevin dynamics [47]. Periodic boundary conditions were adopted. Short-range electrostatic and van der Waals interactions were smoothly truncated with a cutoff of 12 Å, and a switching function was used at 10 Å. Long-range electrostatic interactions were estimated using the particle mesh Ewald algorithm [48,49]. Finally, the systems were simulated in NPT ensemble for 50 ns with a time step of 2 fs, which is named as production simulation. The temperature was also controlled at 300 K using Langevin dynamics [47]. Electrostatic and van der Waals interactions were calculated using the same algorithms in the equilibrium simulation process. The pressure was maintained at 1 atm using the Langevin piston method [50].

### 2.4. Steered Molecular Dynamics (SMD) Simulation

In order to further investigate the influence of E_2_ on the hMTP complex, steered molecular dynamics (SMD) simulations were employed to simulate the disassociation process between hMTPα subunit and hPDI in hMTP and hMTP–E_2_ complexes, and the binding strengths in these two complexes were calculated.

The final conformations in the four MD trajectories were used as the initial structures of SMD simulations. In hMTP and hMTP–E_2_ complexes, all the Cα atoms of hPDI were fixed, and all the Cα atoms of hMTPα subunit were set to the SMD atoms. No hydrogen bond constraint in hMTP and hMTP–E_2_ complexes was set to restrict the distance between hMTPα subunit and hPDI (or E_2_). Extra bond constraints were set to avoid the structural deformation of hMTPα subunit during the SMD simulation process. If the distance between any two Cα atoms of hMTPα was ≤7 Å in the initial structures, the two Cα atoms would be connected using a spring with a spring constant of 10 kcal/mol/Å^2^. Velocity (SMDVel) and spring constant (SMDk) of all the SMD simulations were set to 10 Å/ns and 10 kcal/mol/Å^2^, respectively. Simulation time was set to 2 ns according to the distance changes between hPDI and hMTPα subunit during the SMD simulation process. The SMD direction was selected according to the mean square fluctuations of distance (the definition will be described later) of the interface contacts in hMTP and hMTP–E_2_ complexes during the production simulation process. In order to calculate the potential of mean force (PMF, the calculation method will be described later), SMD simulations were conducted with the same initial conditions 10 times using NAMD [46]. Finally, there were 40 SMD trajectories generated to analyze the binding strengths between hMTPα subunit and hPDI in hMTP and hMTP–E_2_ complexes.

### 2.5. Metrics for Analysis of The MD and SMD Simulation Results

#### 2.5.1. Number of Interface Atom Pairs and Areas of Interface Regions

The number of interface atom pairs can reflect the interface stability to a certain extent. In fact, an earlier study has shown that the network of interface inter-residue contacts (ICs) is a good descriptor for the protein-protein binding affinity [51]. In this work, two atoms from different partners would be regarded as an interface atom pair if the distance between them was ≤5 Å. The number of the interface atom pairs was counted in the conformations of hMTP and hMTP–E_2_ complexes after MD simulations. Additionally, interface areas are also directly correlated to the protein-protein binding affinity [52]. Here, the interface area was calculated using Q_contacts_, which is a Voronoi polyhedra-based method for computing the contact area between different atoms in protein monomer and protein-protein complexes [53].

#### 2.5.2. Mean Square Fluctuation of Distance (MSFD)

The stability of the interface contact between the *i*th and *j*th residues was estimated from molecular dynamics (MD) simulation using mean square fluctuation of the distance between them, which is calculated using the following equation:(1)$$\langle \Delta R_{ij}^{2}\rangle =\langle {\left(R_{ij}-\langle R_{ij}\rangle \right)}^{2}\rangle \approx \frac{\sum_{k=1}^{N}{\left(R_{ij,k}-\langle R_{ij}\rangle \right)}^{2}}{N}$$

Here, $\langle \cdot \rangle$ represents the mean value of variable “·”. *R_ij_* is the distance between the Cα atoms of the *i*th and *j*th residues. $\langle R_{ij}\rangle$ is the mean value of *R_ij_* during the simulation process. In the real environment, there exists infinite conformations for protein because of its flexibility. However, in the simulation process, only finite conformations can be retained. Hence, the mean square fluctuation of distance is approximated based on the finite representative conformations. *N* is the number of representative conformations along the MD trajectory.

#### 2.5.3. Potential of Mean Force (PMF)

The binding free energy between two components in a complex was estimated from steered molecular dynamics (SMD) simulation using Jarzynski’s equation [54], which is expressed below:(2)$$\langle e^{-\beta W}\rangle =e^{-\beta \Delta F}$$

Here, $\langle \cdot \rangle$ is the mean value of variable “·”. β ≡ 1/*k_B_T*, *k_B_* and *T* are Boltzmann constant and the absolute temperature, respectively. *W* and ∆*F* constitute the total work conducted by the SMD force and the free energy difference between the initial and final conformations in the SMD simulation process, respectively. Hence, the binding free energy was calculated using the following equation:(3)$$\Delta F=\frac{\ln(\langle e^{-\beta W}\rangle )}{\beta }\approx \langle W\rangle -\frac{\beta (\langle W^{2}\rangle -{\langle W\rangle }^{2})}{2}$$

In the above equation, the binding free energy is approximated using the Taylor series, and only the first two terms are retained [54]. In order to compute the mean value of *e^−βW^*, *W* and *W*^2^, SMD simulations with the same initial conditions should be conducted at least 10 times [55].

## 3. Results and Discussion

### 3.1. Structures of hMTP and hMTP–E_2_ Complexes

#### 3.1.1. Structure of the hMTP Complex

The complete hMTP complex was generated according to the earlier experimentally determined structure [27] by using the SWISS-MODEL method [30], and the missing residues of the hMTPα subunit and hPDI were added back to their structures. As shown in Figure 1A, the interface residues of hPDI in the hMTP complex are located at the *a*, *b*’ and *a*’ domains. There exists a hydrogen bond between hPDI HIS256 and hMTPα TYR605. Because of flexibility, it is speculated that the missing residues at the C-terminals of hMTPα subunit (residues 886-894) and hPDI (residues 480-508) likely would also make interactions with hPDI and hMTPα subunit, respectively.

#### 3.1.2. Structures of hPDI–E_2_ and hMTP–E_2_ Complexes

The hPDI–E_2_ complex was predicted based on the experimental information on the E_2_ binding site of hPDI using the protein-ligand docking method. Schrodinger Glide-XP (extra precision) [36] was employed to generate the docking conformations of hPDI and E_2_ [36,37,38], and the best binding pose was selected. The poses with the lowest energy were used to sample the nearby torsional minima by the Monte Carlo (MC) procedure. Then, the best poses for hPDI–E_2_ binding were selected according to the GlideScore [39], which includes scoring terms related to buried polar groups and steric clashes. The predicted structure of the hPDI–E_2_ complex is shown in Figure 1B. E_2_ makes contacts with hPDI’s *b*’ domain, with a hydrogen bond formed between the C-3-hydroxyl group of E_2_ and hPDI HIS256. This conformation is slightly different from the one that we reported earlier, due to the different hPDI protein structures used in the docking [28].

The hMTP–E_2_ complex was constructed by the superimposition of the hPDI subunit (taken from the known hMTP complex) onto the hPDI–E_2_ complex, and then the hMTPα subunit (also taken from the known hMTP complex) was placed in the corresponding position next to hPDI–E_2_ complex using VMD [40]. As shown in Figure 1C, the interface structure between hPDI’s *b*’ domain and hMTPα subunit in hMTP–E_2_ complex becomes less compact compared with that in the hMTP complex (Figure 1A), which is caused by the presence of E_2_.

### 3.2. MD Simulations Analysis

MD simulations were conducted for the hMTP and hMTP–E_2_ complexes with/without a hydrogen bond constraint. In the hMTP/hMTP–E_2_ complexes, the hydrogen bond is present between hPDI HIS256 and hMTPα TYR605 and between hPDI and E_2_, respectively (Figure 1A,C). As described in detail below, the MD simulation process incorporates three parts: energy minimization for 10,000 steps, equilibrium simulation in NVT ensemble for 1 ns, and production simulation in NPT ensemble for 50 ns.

#### 3.2.1. Energy Changes during MD Simulations

Energy changes can indirectly reflect the stability of the whole system during the simulation process from an energy perspective. As an example, energy changes during the MD simulation process for the hMTP complex with a hydrogen bond were analyzed to illustrate the stability of the whole system. In the process of energy minimization, the system energy decreases dramatically at first (left and middle panels of Appendix A), then fluctuates around a stable level (right panel of Appendix A), which indicates that the structure was already optimized in terms of energy. In the process of equilibrium and production simulations (Appendix A), the system energy increases firstly, and then fluctuates around a stable level, which reflects that the structure is in a dynamic equilibrium state. The system energy changes during other MD simulation process are similar to the one shown in Appendix A. The structures were optimized in the process of energy minimization as much as possible, and reached dynamic balance states during equilibrium and production simulations.

#### 3.2.2. Conformations after MD Simulations

There are four conformations after MD simulations (MDs) of the hMTP and hMTP–E_2_ complexes with/without a hydrogen bond constraint, which are shown in Figure 2. In order to analyze the binding interfaces in four conformations of hMTP and hMTP–E_2_ complexes, the number of interface atom pairs was counted, and the interface areas were calculated using Q_contacts_ [53]. For convenience, MD simulations with and without a hydrogen bond constraint are named as the first and second MDs, respectively.

After MDs, the overall shapes of hMTP and hMTP–E_2_ complexes (shown in Figure 2A–D) are not changed, and the ranges of the total number of interface atom pairs and the total interface areas between hPDI and hMTPα subunit are 6200~7900 and 3000~3600 Å^2^, respectively (Table 1 and Table 2). Because of the flexibility of the *C*-terminal (residues 480-508) of hPDI, the number of interface atom pairs and interface areas involving the *C*-terminal of hPDI may be very large for the dynamic conformations of hMTP and hMTP–E_2_ complexes under physiologically relevant biochemical environment. When hPDI’s *C*-terminal is not included, the number of interface atom pairs are 6695 vs. 4990 in hMTP and hMTP–E_2_ complexes after the first MDs, and 5389 vs. 4495 in hMTP and hMTP–E_2_ complexes after the second MDs, respectively; areas of the interface regions are 2907.06 Å^2^ and 2347.69 Å^2^, respectively, for hMTP and hMTP–E_2_ complexesafter the first MDs, and 2247.81 Å^2^ and 2223.92 Å^2^, respectively, for hMTP and hMTP–E_2_ complexes after the second MDs. Therefore, the numbers of interface atoms and areas of interface regions in the hMTP complex are higher than the ones in the hMTP–E_2_ complex when hPDI’s *C*-terminal is not considered, which indicates that E_2_ may make the interface between hPDI and hMTPα subunit more flexible (i.e., less stable).

The hydrogen bonds between E_2_, hPDI HIS256 and hMTPα TYR605 in the four conformations after MDs are shown in Figure 2A–D, respectively. When the hydrogen bond between hPDI HIS256 and hMTPα TYR605 (Figure 2A) or between hPDI HIS256 and E_2_ (Figure 2B) was constrained in the first MDs, they would exist in the conformations along the whole trajectories, which include the final conformations, as shown in Figure 2A,B. In the conformation of the hMTP–E_2_ complex (Figure 2B), there is another hydrogen bond formed between E_2_ and hMTPα-TYR605, which increases the interface stability between hPDI and hMTPα subunit. After the second MDs, the hydrogen bond between hPDI HIS256 and hMTPα TYR605 still exists in the conformation of the hMTP complex (Figure 2C), which indicates that the hydrogen bond is relatively stable. In the conformation of the hMTP–E_2_ complex after the second MDs (Figure 2D), there are two hydrogen bonds formed between E_2_ and hPDI HIS256/hMTPα TYR605. The relative positions between E_2_, hPDI HIS256 and hMTPα TYR605 (Figure 2D) are different from the ones in the conformation of the hMTP–E_2_ complex after the first MDs (Figure 2B). Without the hydrogen bond constraint in the second MDs, E_2_ rotates about 180 degrees after MD simulations, which illustrates that the hMTPα subunit may affect the binding pose between hPDI and E_2_.

The number of interface atom pairs between E_2_ and hPDI or hMTPα subunit was also counted in the conformations of hMTP–E_2_ complexes after MDs (Appendix A). The total numbers of interface atom pairs between E_2_ and hPDI or hMTPα subunit are 1014 vs. 1053 in the conformations of hMTP–E_2_ complexes after the first vs. second MDs, respectively. Although the total numbers are close to each other, the numbers of interface atom pairs between E_2_ and hPDI in hMTP–E_2_ complex after the first MDs is lower than the one in the hMTP–E_2_ complex after the second MDs (694 vs. 852); the number of interface atom pairs between E_2_ and hMTPα subunit in the hMTP–E_2_ complex after the first MDs is higher than the one in hMTP–E_2_ complex after the second MDs (320 vs. 201). Rotation of E_2_ increases its interactions with hPDI, and decreases its interactions with hMTPα subunit. Regardless of the rotation of E_2_, the number of interface atom pairs between E_2_ and hPDI is higher than the one between E_2_ and hMTPα subunit (694 vs. 320 and 852 vs. 201), which indicates that the interactions in the former are stronger than the ones in the latter.

#### 3.2.3. Mean Square Fluctuation of Distance (MSFD) of the Interface Contacts during MD Simulations

In order to illustrate the interface stability of hMTP and hMTP–E_2_ complexes during the MD simulation process, the mean square fluctuation of distance (MSFD) of the interface contacts between different residues from hPDI and hMTPα subunit were calculated using Equation (1) based on more than 350 representative conformations retained and selected from the MD trajectories. If the distance between the Cα atoms of two residues from hPDI and hMTPα subunit was equal to or smaller than 7 Å in any one conformation along the trajectory, the two residues would be regarded as an interface contact. The four MD trajectories have 71 shared interface contacts, and there are 43 interface contacts between hPDI and hMTPα subunit in the crystal structure of the hMTP complex [27]. After integrating the interface information based on the MD trajectories with the information from the crystal structure [27], it appears that there are a total of 93 interface contacts between hPDI and hMTPα subunit and they can be categorized into four regions (Figure 3A): region 1 (23 interface contacts between hPDI a domain and hMTPα subunit), region 2 (31 interface contacts between hPDI *b*’ domain and hMTPα subunit), region 3 (31 interface contacts between hPDI *a*’ domain and hMTPα subunit), region 3′ (8 interface contacts between hPDI *C*-terminal and hMTPα subunit).

The high MSFD reflects a high degree of flexibility (or instability) of the interface contacts. In the interface regions between hPDI and hMTPα subunit in hMTP and hMTP–E_2_ complexes, the MSFDs of 8 interface contacts involving the residues at the *C*-terminal of hMTPα subunit are higher than 14 Å^2^ in at least one of the four trajectories (Appendix A), which may be caused by the flexibility of the *C*-terminal of hMTPα subunit. The MSFDs of the remaining 85 interface contacts are shown in Figure 3B,C. With a hydrogen bond constraint in the MD simulation process, the MSFDs of interface contacts in hMTP and hMTP–E_2_ complexes have some similarities and differences (Figure 3B). There exist small differences in region 1 as well as part of region 2; the MSFDs of interface contacts in part of region 2 and most of region 3′ of hMTP complex are higher than those of the hMTP–E_2_ complex; the MSFDs of interface contacts in region 3 of the hMTP complex are lower than those of the hMTP–E_2_ complex. Overall, there are 25 interface contacts, for which the MSFDs in the hMTP complex are 1 Å^2^ higher than those in the hMTP–E_2_ complex, and 40 interface contacts, for which the MSFDs in the hMTP–E_2_ complex are 1 Å^2^ higher than those in the hMTP complex (Appendix A), which indicates that E_2_ makes the interface between the hMTPα subunit and hPDI more flexible.

When the hydrogen bond constraint is absent during the simulations, the MSFDs of interface contacts are shown in Figure 3C. The MSFDs of interface contacts in part of region 1 and 3′ of hMTP complex are higher than those of hMTP–E_2_ complex; the MSFDs of interface contacts in region 2 and most of region 3 of hMTP complex are lower than those of hMTP–E_2_ complex. Overall, there are 26 interface contacts for which the MSFDs in hMTP complex are 1 Å^2^ higher than those in hMTP–E_2_ complex, and 42 interface contacts for which the MSFDs in hMTP–E_2_ complex are 1 Å^2^ higher than those in hMTP complex (Appendix A), which indicate that E_2_ can make the interface between hMTPα subunit and hPDI more flexible.

In order to further compare the overall stabilities in different interface regions of hMTP and hMTP–E_2_ complexes, the average of MSFDs in different interface regions were calculated based on the MSFDs of interface contacts shown in Figure 3B,C. The results are shown in Figure 4 and Appendix A. High averages of MSFDs reflect the high flexibility (or instability) of the interface regions. With a hydrogen bond constraint present during the simulations (Figure 4A and Appendix A), absolute values of the differences between the averages of MSFDs in hMTP and hMTP–E_2_ complexes are smaller than 1 Å^2^ in the interface regions 1 and 2, which indicates that the influence of E_2_ in these regions is not apparent. The average MSFD in interface region 3 of hMTP complex is higher than the one of hMTP–E_2_ complex (1.28 Å^2^ vs. 3.20 Å^2^), which implies that interface region 3 of the hMTP complex is less flexible than the one in the hMTP–E_2_ complex.

In the absence of a hydrogen bond constraint during the simulations (Figure 4B and Appendix A), the relationship of the average MSFDs between hMTP and hMTP–E_2_ complexes is as follows: in the interface region 1, average of MSFDs in hMTP complex > average of MSFDs in hMTP–E_2_ complex (5.29 Å^2^ vs. 3.01 Å^2^); in the interface region 2 and 3, the average of MSFDs in hMTP complex is smaller than the average of MSFDs in hMTP–E_2_ complex (0.60 Å^2^ vs. 3.70 Å^2^ and 2.08 Å^2^ vs. 3.61 Å^2^). These results reflect that E_2_ makes the interface region 2 and 3 more flexible, but makes the interface region 1 more stable, which may be related to the rotation of E_2_ during MD simulations. Regardless of the hydrogen bond constraint, the average of MSFDs in region 3′ of hMTP complex is higher than the one of hMTP–E_2_ complex (3.96 Å^2^ vs. 2.38 Å^2^ and 3.53 Å^2^ vs. 2.17 Å^2^), which may be related to the relative positions between the *C*-terminal of hPDI and hMTPα subunit in the initial structures of hMTP and hMTP–E_2_ complexes before MD simulations. The average of MSFDs in the four interface regions of the hMTP complex is lower than the one of the hMTP–E_2_ complex (2.43 Å^2^ vs. 2.72 Å^2^ and 2.61 Å^2^ vs. 3.34 Å^2^). Overall, these results all indicate that E_2_ would make the interface of hMTP complex more flexible (or less stable).

### 3.3. SMD Simulation Analysis

SMD simulations were performed to investigate the binding strengths between hPDI and hMTPα subunit in hMTP and hMTP–E_2_ complexes. The SMD direction was selected based on the interface regions with the highest average of MSFDs (Appendix A and Figure 4). Without consideration of the interface region involving *C*-terminal of hPDI (i.e., region 3′), the two interface regions with the highest average of MSFDs are region 1 of the hMTP complex (5.29 Å^2^) and region 2 of the hMTP–E_2_ complex (3.70 Å^2^) when a hydrogen bond constraint is absent during the simulations. It may be appropriate to select the SMD direction in the midpoint region between interface regions 1 and 2 (Figure 3A). The position and direction of a helix (residues 591-595) of the hMTPα subunit are approximate to the midpoint region of two interface regions, and the helix was regarded as a reference of the SMD direction. Directions of all the SMD simulations are from the Cα atom of hMTPα ARG595 to the Cα atom of residue LYS591. Ten trajectories for each conformation were generated with the same initial conditions. In total, there are forty SMD trajectories for the four conformations of hMTP and hMTP–E_2_ complexes after MD simulations. For convenience of description, SMD simulations starting from the conformations after MDs with and without a hydrogen bond constraint are named as the first and second SMDs, respectively.

#### 3.3.1. Representative Conformations, Changes of SMD Forces and Distances between hPDI and hMTPα Subunit during SMD Simulations

In order to visualize the conformational changes of hPDI and hMTP–E_2_ complexes during SMD simulations, representative conformations were extracted from the SMD trajectories (Figure 5). The interactions between hPDI and hMTPα subunit are strong in the initial conformations, so the distances between them increase slowly at the earlier stages (0~0.5 ns) during the SMD simulations (Figure 5). With the increase of the SMD forces, the distances between hPDI and hMTPα subunit show apparent increases from 0.5 ns to 1 ns. After the dissociation of hMTPα subunit, the interactions become weak, which causes the distances between the two protein subunits to increase rapidly in the later stage (1.5~2.0 ns) of the SMD simulations (Figure 5). E_2_ dissociates from hPDI in the conformations of hMTP–E_2_ complex at the later period of the second SMDs (Figure 5D). One reason might be that all the Cα atoms of hPDI were fixed and the overall conformation of hPDI was fixed, which is viewed unreasonable under real biochemical environments. E_2_ may induce the changes of the overall conformation of hPDI after the dissociation of hMTPα subunit. Accordingly, the results from the initial phases of SMD simulations are more reliable than the results from the middle and later phases of SMD simulations.

In order to illustrate the differences of the binding strengths between hPDI and hMTPα subunit in hMTP and hMTP–E_2_ complexes, changes of the distance with variations of the SMD forces at the initial phase (the first 0.5 ns) of SMD simulations are shown in Figure 6. The distances between hPDI and hMTPα subunit during SMD simulations are reflected by the change of the midpoint position between the two Cα atoms of the two residues (ARG595 and LYS591) in hMTPα. No distinct difference is observed for the first SMDs results for the hMTP complex, either in the presence or absence of E_2_ (Figure 6A). In comparison, distinct differences are observed for the hMTP and hMTP–E_2_ complexes during the second SMDs processes (Figure 6B). With the same SMD forces for the hMTP and hMTP–E_2_ complexes, the distances between hMTPα subunit and hPDI in the hMTP–E_2_ complex are longer than the ones in the hMTP complex, which indicates that the interaction between hMTPα subunit and hPDI in hMTP–E_2_ complex is weaker than the one in the hMTP complex.

#### 3.3.2. Calculation of the Potential of Mean Force (PMF)

PMFs during the SMD simulation process were calculated using Equation (3) based on the SMD trajectories. According to the analysis of the force-distance plots as shown in Figure 6, there is no apparent difference between the first SMDs for hMTP and hMTP–E_2_ complexes (Figure 6A), and the corresponding PMFs are not shown here. The PMFs shown in Figure 7 were derived from the first 1 ns of the second SMDs for the hMTP and hMTP–E_2_ complexes. The PMFs for the hMTP complex are higher than the ones for the hMTP–E_2_ complex (Figure 7), which indicates that interactions between hPDI and hMTPα subunit in the hMTP complex are stronger than the ones in the hMTP–E_2_ complex.

## 4. Conclusions

In the present study, the influence of E_2_ on the stability of the hMTP complex is investigated using molecular modelling analyses based on available experimental information. The results of MD and SMD simulations demonstrate that E_2_ may make the interface between hPDI and the hMTPα subunit more flexible and thus less stable. It appears that the presence of hMTPα subunit affects the binding pose between E_2_ and hPDI. Since hMTP is essential for the assembly of apoB-containing lipoproteins, the new knowledge gained from computational modelling analyses of E_2_-induced conformational changes in the hMTP complex might offer new clues on the ER-independent lipid-modulating effect of E_2_.

## Figures and Tables

**Figure 1 cells-10-01566-f001:**
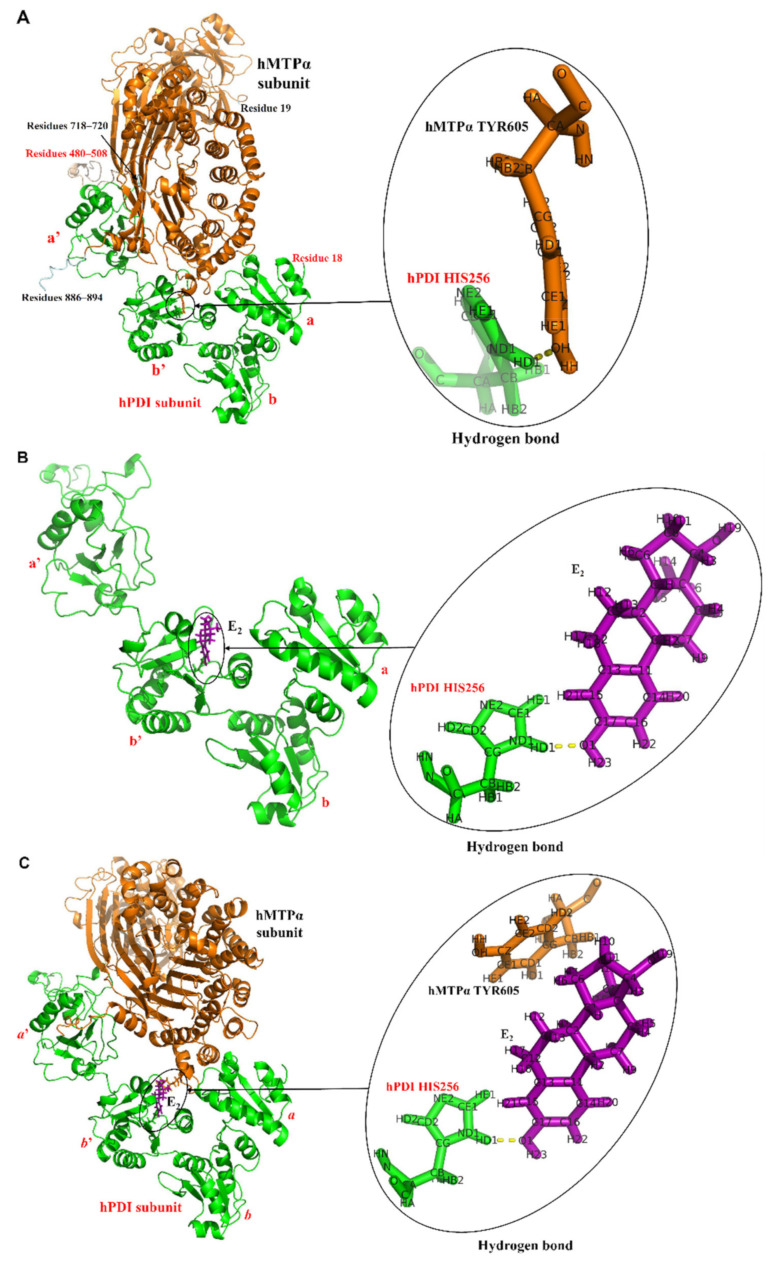
Complete structures of hMTP, hPDI–E_2_ and hMTP–E_2_ complexes. (**A**) Complete structure of the hMTP complex by adding the missing residues of hMTPα subunit (13 residues) and hPDI (30 residues) back in their experimentally determined structures. There is a hydrogen bond formed between hPDI HIS256 and hMTPα TYR605. (**B**) Structure of the hPDI–E_2_ complex predicted using the protein-ligand docking approach. A key hydrogen bond exists between E_2_ and hPDI HIS256. (**C**) Structure of the hMTP–E_2_ complex constructed using structural alignment. The hMTPα subunit, hPDI and E_2_, are colored in orange, green and purple, respectively. The missing residues in hMTPα subunit and hPDI are colored in wheat and palecyan, respectively.

**Figure 2 cells-10-01566-f002:**
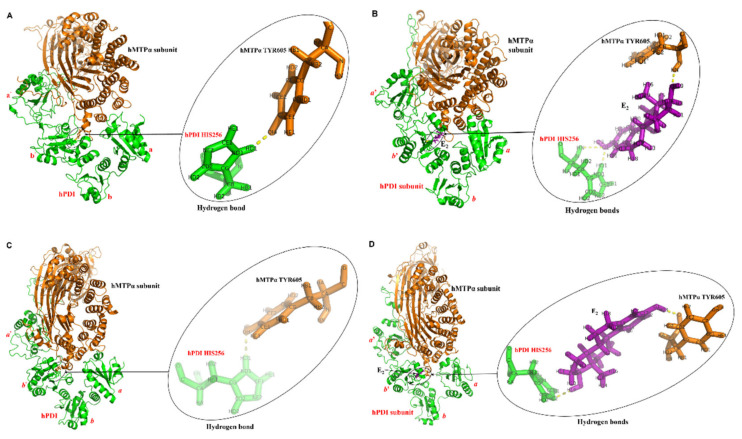
Conformations of hMTP and hMTP–E_2_ complexes after molecular dynamics (MD) simulations. For convenience, MD simulations with and without a hydrogen bond constraint are called as the first and second MDs, respectively. (**A**) hMTP complex after the first MDs. (**B**) hMTP–E_2_ complex after the first MDs. (**C**) hMTP complex after the second MDs. (**D**) hMTP–E_2_ complex after the second MDs. The hMTPα subunit, hPDI and E_2_ are colored in orange, green and purple, respectively.

**Figure 3 cells-10-01566-f003:**
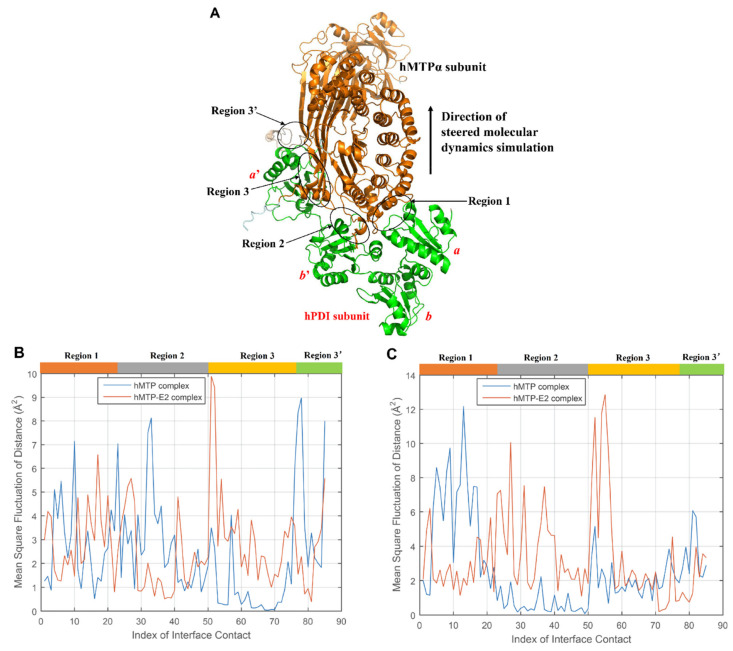
Mean square fluctuation of distance (MSFD) of the interface contacts during the production simulation process. (**A**) Different interface regions between hMTPα subunit and hPDI. (**B**) MSFDs of the interface contacts in hMTP and hMTP–E_2_ complexes with a hydrogen bond constraint in the simulation process. (**C**) MSFDs of the interface contacts in hMTP and hMTP–E_2_ complexes without the hydrogen bond constraint in the simulation process.

**Figure 4 cells-10-01566-f004:**
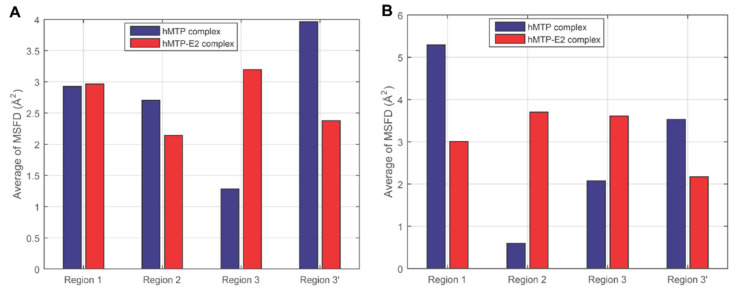
Average of mean square fluctuation of distance (MSFD) of the interface contacts during the production simulation process. (**A**) Average of MSFDs in the four interface regions of hMTP and hMTP–E_2_ complexes with a hydrogen bond constraint in the simulation process. (**B**) Average of MSFDs in the four interface regions of hMTP and hMTP–E_2_ complexes without the hydrogen bond constraint in the simulation process.

**Figure 5 cells-10-01566-f005:**
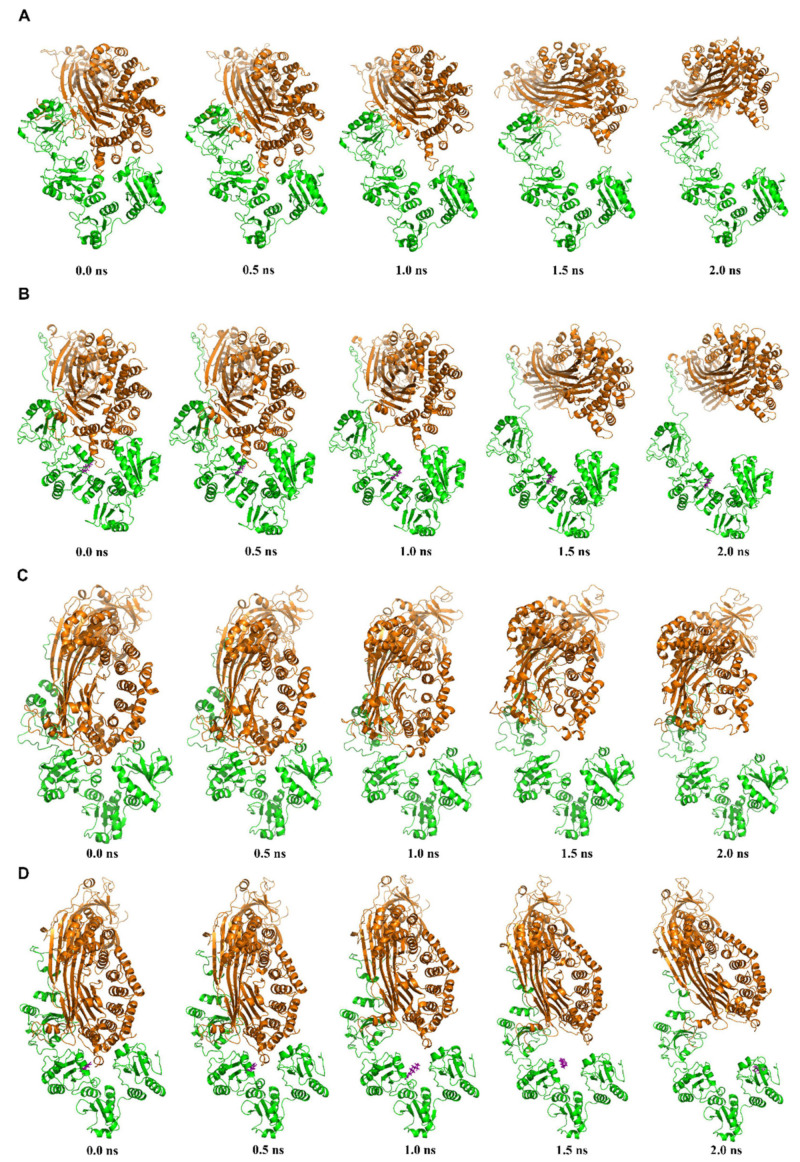
Representative conformations of hMTP and hMTP–E_2_ complexes during the steered molecular dynamics (SMD) simulation process. For convenience of description, SMD simulations using the conformations after MDs with and without a hydrogen bond constraint are named as the first and second SMDs, respectively. (**A**) Representative snapshots of hMTP complex during the first SMDs. (**B**) Representative snapshots of hMTP–E_2_ complex during the first SMDs. (**C**) Representative snapshots of hMTP complex during the second SMDs. (**D**) Representative snapshots of hMTP–E_2_ complex during the second SMDs. The hMTPα subunit, hPDI and E_2_ are colored in orange, green and purple, respectively.

**Figure 6 cells-10-01566-f006:**
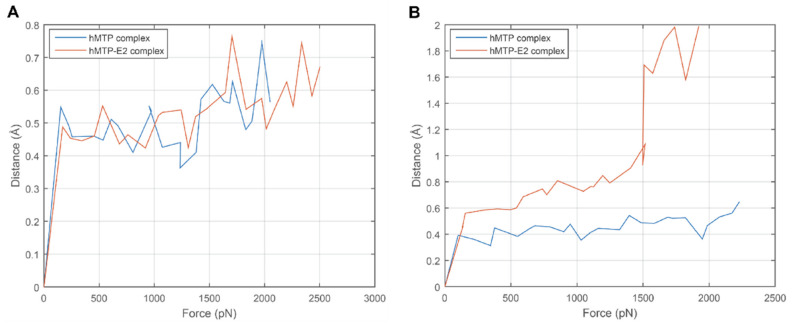
Changes of distances between hMTPα subunit and hPDI with the SMD forces during the SMD simulation process at the initial stage (the first 0.5 ns). For convenience of description, SMD simulations using the conformations after MDs with and without a hydrogen bond constraint are called the first and second SMDs, respectively. (**A**) Changes of distances between hMTPα subunit and hPDI with the SMD forces during the first SMDs. (**B**) Changes of distances between hMTPα subunit and hPDI with the SMD forces during the second SMDs. The SMD forces and distances shown here are the averages of the SMD forces and distances between hMTPα subunit and hPDI in the ten SMD trajectories for each complex, respectively.

**Figure 7 cells-10-01566-f007:**
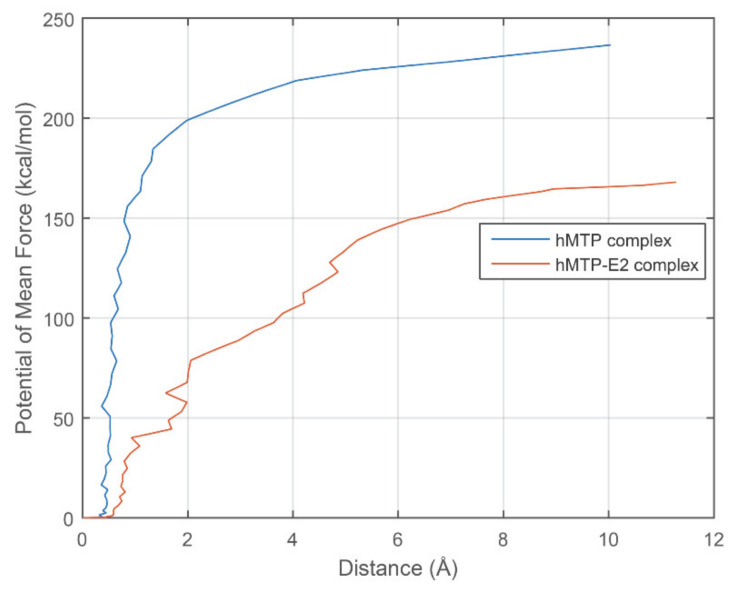
Potential of mean force (PMF) derived from the steered molecular dynamics (SMD) simulation process of the first 1 ns. PMFs of hMTP and hMTP–E_2_ complexes calculated from the SMD simulations which start from the conformation after the MD simulations without the hydrogen bond constraint.

**Table 1 cells-10-01566-t001:** Number of the interface atom pairs between hPDI and hMTPα subunit in the four conformations of hMTP and hMTP–E_2_ complexes after MD simulations.

Interacting Region in hPDI	Number of the Interface Atom Pairs (cutoff = 5 Å)			
	with a Hydrogen Bond Constraint in the Simulation		without the Hydrogen Bond Constraint in the Simulation	
	hMTP Complex	hMTP–E_2_ Complex	hMTP Complex	hMTP–E_2_ Complex
*N*-terminal (residues 18–25)	52	0	0	0
*a* domain	2126	1802	1588	1117
*b* domain	117	0	0	0
*b*’ domain	1644	1643	2483	1298
x (residues 350–368)	0	38	0	54
*a*’ domain	2756	1507	1318	2026
*C*-terminal (residues 480–508)	838	2880	2082	1793
All	7533	7870	7471	6288
All except *C*-terminal	6695	4990	5389	4495

**Table 2 cells-10-01566-t002:** Area of the interface regions in the four conformations of hMTP and hMTP–E_2_ complexes after MD simulations.

Interacting Region in hPDI	Area of the Interface Regions (Å^2^)			
	with a Hydrogen Bond Constraint in the Simulation		without the Hydrogen Bond Constraint in the Simulation	
	hMTP Complex	hMTP–E_2_ Complex	hMTP Complex	hMTP–E_2_ Complex
*N*-terminal (residues 18–25)	43.49	0	0	0
*a* domain	849.08	839.51	699.27	560.53
*b* domain	106.84	0	0	0
*b*’ domain	677.75	765.37	912.12	720.47
x (residues 350–368)	0	30.17	0	20.71
*a*’ domain	1229.90	712.65	636.42	922.22
*C*-terminal (residues 480–508)	392.16	1246.01	1059.37	848.93
All	3299.22	3593.70	3307.18	3072.85
All except *C*-terminal	2907.06	2347.69	2247.81	2223.92

## Data Availability

The data presented in this study are available on request from the corresponding author.

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
