# Peer review of "17β-Estradiol-Induced Conformational Changes of Human Microsomal Triglyceride Transfer Protein: A Computational Molecular Modelling Study"

_cells, 2021, doi:10.3390/cells10071566_

Round 1

Reviewer 1 Report

The manuscript by Yang at al. describes molecular modelling of 17beta estradiol hormone (E2) interactions with hMTP.  The computational studies  show that stability of the hMTP heterodimer is significantly reduced in the presence of E2, which is in line with previous reports of inhibition of hPDI function by the female sex hormone and support earlier studies of the corresponding author showing that E2 alters lipid metabolism in the ER-independent manner. The introduction is clearly presented and methods used for protein modelling are appropriate. Conclusions seem .

While the manuscript seems to be written in acceptable English, inappropriate use of Present tense everywhere throughout the Materials and Methods and Results and Discussion makes the manuscript difficult to read and comprehend. The authors should change the tense to Past Indefinite or Past Perfect almost everywhere throughout the text.

As an example LL 102-104

The structure ... are downloaded --> ... was downloaded or ... has been downloaded

which is used as a template --> this was used 

Author Response

RESPONSE: The tenses have been changed throughout the text. The revisions are described as follows:

  1. Line 100 in the original version: “are added” is changed to “were added” at line 100 in the revised version.
  2. Line 102 in the original version: “are downloaded” is replaced by “were downloaded” at line 102 in the revised version.
  3. Line 103 in the original version: “are downloaded” is replaced by “was downloaded” at line 103 in the revised version.
  4. Line 103 in the original version: “is used” is replaced by “was used” at line 103-104 in the revised version.
  5. Line 107 in the original version: “is predicted” is changed to “was predicted” at line 107 in the revised version.
  6. Line 109 in the original version: “is processed” is replaced by “was processed” at line 109 in the revised version.
  7. Line 111 in the original version: “are added” is changed to “were added” at line 111 in the revised version.
  8. Line 112 in the original version: “are adjusted” is replaced by “were adjusted” at line 112 in the revised version.
  9. Line 113 in the original version: “are added” is changed to “were added” at line 113 in the revised version.
  10. Line 114 in the original version: “are deleted” is replaced by “were deleted” at line 114 in the revised version.
  11. Line 114-115 in the original version: “are subjected” is changed to “were subjected” at line 114-115 in the revised version.
  12. Line 117-118 in the original version: “is used” is replaced by “was used” at line 117-118 in the revised version.
  13. Line 121 in the original version: “are subjected” is changed to “were subjected” at line 121 in the revised version.
  14. Line 123 in the original version: “are determined” is replaced by “were determined” at line 123 in the revised version.
  15. Line 124 in the original version: “are generated” is replaced by “were generated at line 124 in the revised version.
  16. Line 125 in the original version: “is centered” is changed to “was centered” at line 125 in the revised version.
  17. Line 126 in the original version: “are set” is replaced by “were set” at line 126 in the revised version.
  18. Line 128 in the original version: “is constructed” is replaced by “was constructed” at line 128 in the revised version.
  19. Line 129 in the original version: “is placed” is replaced by “was placed” at line 129 in the revised version.
  20. Line 131 in the original version: “is further optimized” is replaced by “was further optimized” at line 131 in the revised version.
  21. Line 134 in the original version: “are conducted” is replaced by “were conducted” at line 134 in the revised version.
  22. Line 138 in the original version: “are extracted” is replaced by “were extracted” at line 138 in the revised version.
  23. Line 138 in the original version: “are added” is replaced by “were added” at line 138-139 in the revised version.
  24. Line 140 in the original version: “is also used” is replaced by “was also used” at line 140 in the revised version.
  25. Line 141 in the original version: “are calculated” is replaced by “were calculated” at line 141 in the revised version.
  26. Line 141 in the original version: “is adopted” is replaced by “was adopted for protein” at line 142 in the revised version.
  27. Line 142 in the original version: “are embedded” is replaced by “were embedded” at line 142 in the revised version.
  28. Line 143 in the original version: “is used” is replaced by “was used” at line 143-144 in the revised version.
  29. Line 144 in the original version: “are added” is replaced by “were added” at line 144 in the revised version.
  30. Line 149 in the original version: “are conducted” is replaced by “were conducted” at line 149 in the revised version.
  31. Line 155 in the original version: “are energy minimized” is replaced by “were energy minimized” at line 155 in the revised version.
  32. Line 156 in the original version: “are equilibrated” is replaced by “were equilibrated” at line 156 in the revised version.
  33. Line 157 in the original version: “is set” is replaced by “was set” at line 157 in the revised version.
  34. Line 157-158 in the original version: “is maintained” is replaced by “was maintained” at line 158 in the revised version.
  35. Line 158-159 in the original version: “are adopted” is replaced by “were adopted” at line 159 in the revised version.
  36. Line 159 in the original version: “are smoothly” is replaced by “were smoothly” at line 159 in the revised version.
  37. Line 160 in the original version: “is used” is replaced by “was used” at line 160 in the revised version.
  38. Line 161 in the original version: “are estimated” is replaced by “were estimated” at line 161 in the revised version.
  39. Line 162 in the original version: “are simulated” is replaced by “were simulated” at line 162 in the revised version.
  40. Line 163 in the original version: “is also” is replaced by “was also” at line 163 in the revised version.
  41. Line 164 in the original version: “are calculated” is replaced by “were calculated” at line 164-165 in the revised version.
  42. Line 165 in the original version: “is maintained” is replaced by “was maintained” at line 165-166 in the revised version.
  43. Line 169 in the original version: “are employed” is replaced by “were employed” at line 169 in the revised version.
  44. Line 171 in the original version: “are calculated” is replaced by “were calculated” at line 171 in the revised version.
  45. Line 172 in the original version: “are used” is replaced by “were used” at line 172 in the revised version.
  46. Line 173-174 in the original version: “are fixed” is replaced by “were fixed” at line 174 in the revised version.
  47. Line 174 in the original version: “are set” is replaced by “were set” at line 174 in the revised version.
  48. Line 175 in the original version: “is set” is replaced by “was set” at line 175 in the revised version.
  49. Line 176 in the original version: “are set” is replaced by “were set” at line 176 in the revised version.
  50. Line 178 in the original version: “is 7 Å” is replaced by “was 7 Å” at line 178 in the revised version.
  51. Line 180 in the original version: “are set” is replaced by “were set” at line 180 in the revised version.
  52. Line 181 in the original version: “is set” is replaced by “was set” at line 181 in the revised version.
  53. Line 183 in the original version: “is selected” is replaced by “was selected” at line 183 in the revised version.
  54. Line 184 in the original version: “is described” is replaced by “will be described” at line 184 in the revised version.
  55. Line 186 in the original version: “is described” is replaced by “will be described” at line 186 in the revised version.
  56. Line 186 in the original version: “are conducted” is replaced by “were conducted” at line 186-187 in the revised version.
  57. Line 187-188 in the original version: “are 40 SMD trajectories” is replaced by “were 40 SMD trajectories” at line 188 in the revised version.
  58. Line 191 in the original version: “Number of Interface Atom Pairs and Areas of Interface Regions” is replaced by “2.5.1. Number of Interface Atom Pairs and Areas of Interface Regions” at line 191 in the revised version.
  59. Line 195 in the original version: “are regarded” is replaced by “would be regarded” at line 195 in the revised version.
  60. Line 195 in the original version: “is 5 Å” is replaced by “was 5 Å” at line 196 in the revised version.
  61. Line 196 in the original version: “is counted” is replaced by “was counted” at line 196 in the revised version.
  62. Line 198-199 in the original version: “is calculated” is replaced by “was calculated” at line 199 in the revised version.
  63. Line 202 in the original version: “2.5.1. Mean Square Fluctuation of Distance (MSFD)” is replaced by “2.5.2. Mean Square Fluctuation of Distance (MSFD)” at line 202 in the revised version.
  64. Line 203 in the original version: “is estimated” is replaced by “was estimated” at line 203 in the revised version.
  65. Line 214 in the original version: “2.5.2. Potential of Mean Force (PMF)” is replaced by “2.5.3. Potential of Mean Force (PMF)” at line 214 in the revised version.
  66. Line 215 in the original version: “is estimated” is replaced by “was estimated” at line 215 in the revised version.
  67. Line 222 in the original version: “is calculated” is replaced by “was calculated” at line 222-223 in the revised version.
  68. Line 231 in the original version: “is generated” is replaced by “was generated” at line 231 in the revised version.
  69. Line 234 in the original version: “are added” is replaced by “were added” at line 234 in the revised version.
  70. Line 241 in the original version: “is predicted” is replaced by “was predicted” at line 241 in the revised version.
  71. Line 243 in the original version: “is employed” is replaced by “was employed” at line 243 in the revised version.
  72. Line 244 in the original version: “is selected” is replaced by “was selected” at line 244 in the revised version.
  73. Line 244 in the original version: “are used” is replaced by “were used” at line 244-245 in the revised version.
  74. Line 246 in the original version: “are selected” is replaced by “were selected” at line 246 in the revised version.
  75. Line 248 in the original version: “is show” is replaced by “is shown” at line 248 in the revised version.
  76. Line 251 in the original version: “PDI” is replaced by “hPDI” at line 251 in the revised version.
  77. Line 261 in the original version: “is constructed” is replaced by “was constructed” at line 261 in the revised version.
  78. Line 263 in the original version: “is placed” is replaced by “was placed” at line 263 in the revised version.
  79. Line 269 in the original version: “are conducted” is replaced by “were conducted” at line 269 in the revised version.
  80. Line 278 in the original version: “are analyzed” is replaced by “were analyzed” at line 278 in the revised version.
  81. Line 282 in the original version: “is already optimized” is replaced by “was already optimized” at line 282 in the revised version.
  82. Line 284 in the original version: “is in” is replaced by “was in” at line 284 in the revised version.
  83. Line 286 in the original version: “are optimized” is replaced by “were optimized” at line 286 in the revised version.
  84. Line 287 in the original version: “reach dynamic” is replaced by “reached dynamic” at line 287 in the revised version.
  85. Line 290 in the original version: “MD simulations” is replaced by “MD simulations (MDs)” at line 290 in the revised version.
  86. Line 293 in the original version: “is counted” is replaced by “was counted” at line 293 in the revised version.
  87. Line 293-294 in the original version: “are calculated” is replaced by “were calculated” at line 294 in the revised version.
  88. Line 294 in the original version: One sentence, “For convenience, MD simulations with and without a hydrogen bond constraint are named as the first and second MDs, respectively.”, is added after “Qcontacts [53].” at line 294-295 in the revised version.
  89. Line 295 in the original version: “The overall shapes” is replaced by “After MDs, the overall shapes” at line 296 in the revised version.
  90. Line 296 in the original version: “not changed after MD simulations. The ranges” is replaced by “not changed. In the conformations of hMTP and hMTP–E2 complexes after MDs, the ranges” at line 297-298 in the revised version.
  91. Line 297-298 in the original version: “in four conformations of hMTP and hMTP–E2 complexes after MD simulations” is deleted in the revised version.
  92. Line 303-310 in the original version: “When hPDI’s C-terminal is not included, the number of interface atom pairs are 6695 vs. 4990 for the final conformations of hMTP and hMTP–E2 complexes with a hydrogen bond constraint, and 5389 vs. 4495 for the final conformations of hMTP and hMTP–E2 com-plexes without the hydrogen bond constraint, respectively; areas of the interface regions are 2907.06 Å2 vs. 2347.69 Å2 for the final conformations of hMTP and hMTP–E2 complex-es with a hydrogen bond constraint, 2247.81 Å2 2223.92 Å2 for the final conformations of hMTP and hMTP–E2 complexes without the hydrogen bond constraint, respectively.” is replaced by “When hPDI’s C-terminal is not included, the number of interface atom pairs are 6695 vs. 4990 in hMTP and hMTP–E2 complexes after the first MDs, and 5389 vs. 4495 in hMTP and hMTP–E2 complexes after the second MDs, respectively; areas of the interface regions are 2907.06 Å2 vs. 2347.69 Å2 in hMTP and hMTP–E2 complexes after the first MDs, and 2247.81 Å2 vs. 2223.92 Å2 in hMTP and hMTP–E2 complexes after the second MDs, respectively.” at line 303-309 in the revised version.
  93. Line 311 in the original version: “the conformations of” is deleted in the revised version.
  94. Line 311-312 in the original version: “the conformations of” is deleted in the revised version.
  95. Line 322 in the original version: “conformations are show” is replaced by “conformations after MDs are shown” at line 320 in the revised version.
  96. Line 324 in the original version: “is constrained during MD simulations” is replaced by “was constrained in the first MDs” at line 322 in the revised version.
  97. Line 328-329 in the original version: “Without the hydrogen bond constraint during the MD simulation process” is replaced by “After the second MDs” at line 326 in the revised version.
  98. Line 330-331 in the original version: “after MD simulations” is deleted in the revised version.
  99. Line 332 in the original version: “after the MD simulation process” is replaced by “after the second MDs” at line 329 in the revised version.
  100. Line 335-336 in the original version: “with a hydrogen bond constraint during the MD simulation process” is replaced by “after the first MDs” at line 332 in the revised version.
  101. Line 336-337 in the original version: “Without the hydrogen bond constraint” is replaced by “Without the hydrogen bond constraint in the second MDs” at line 333 in the revised version.
  102. Line 337 in the original version: “after MD simulations” is replaced by “after MDs” at line 334 in the revised version.
  103. Line 340-341 in the original version: “is also counted” is replaced by “was also counted” at line 337-338 in the revised version.
  104. Line 341 in the original version: “the final conformation of the hMTP–E2 complex” is replaced by “the conformations of hMTP–E2 complexes after MDs” at line 338 in the revised version.
  105. Line 343-344 in the original version: “with without the hydrogen bond constraint during the MD simulation process, respectively.” is replaced by “after the first vs. second MDs, respectively.” at line 340 in the revised version.
  106. Line 346 in the original version: “with a hydrogen bond constraint in the MD simulation” is replaced by “after the first MDs” at line 342 in the revised version.
  107. Line 347 in the original version: “without a hydrogen bond constraint” is replaced by “after the second MDs” at line 343 in the revised version.
  108. Line 348-349 in the original version: “between E2 and the hMTPα subunit in the hMTP–E2 complex with a hydrogen bond constraint” is replaced by “between E2 and hMTPα subunit in the hMTP–E2 complex after the first MDs” at line 344 in the revised version.
  109. Line 349-350 in the original version: “without a hydrogen bond constraint” is replaced by “after the second MDs” at line 345 in the revised version.
  110. Line 350-351 in the original version: “with hPDI, decreases” is replaced by “with hPDI, and decreases” at line 346 in the revised version.
  111. Line 358-362 in the original version: “(A) Conformation of hMTP complex with a hydrogen bond constraint after the MD simulation process. (B) Conformation of hMTP–E2 complex with a hydrogen bond constraint after simulation. (C) Conformation of hMTP complex without the hydrogen bond constraint after simulation. (D) Conformation of hMTP–E2 complex without the hydrogen bond constraint after simulation.” is replaced by “For convenience, MD simulations with and without a hydrogen bond constraint are called as the first and second MDs, respectively. (A) hMTP complex after the first MDs. (B) hMTP–E2 complex after the first MDs. (C) hMTP complex after the second MDs. (D) hMTP–E2 complex after the sec-ond MDs.” at line 353-356 in the revised version.
  112. Line 367 in the original version: “are calculated” is replaced by “were calculated” at line 362-363 in the revised version.
  113. Line 370 in the original version: “is equal” is replaced by “was equal” at line 365 in the revised version.
  114. Line 371 in the original version: “are regarded” is replaced by “would be regarded” at line 366 in the revised version.
  115. Line 400 in the original version: “3 of hMTP complex” is replaced by “most of region 3 of hMTP complex” at line 395 in the revised version.
  116. Line 407-408 in the original version: “are calculated” is replaced by “were calculated” at line 402-403 in the revised version.
  117. Line 411 in the original version: “during the simulations” is replaced by “during the simulations (Figure 4A and Table S4)” at line 406 in the revised version.
  118. Line 431 in the original version: “during the simulations” is replaced by “during the simulations (Figure 4B and Table S4)” at line 426-427 in the revised version.
  119. Line 447 in the original version: “are performed” is replaced by “were performed” at line 442 in the revised version.
  120. Line 448 in the original version: “is selected” is replaced by “was selected” at line 443-444 in the revised version.
  121. Line 456 in the original version: “and are regarded” is replaced by “and the helix was regarded” at line 451 in the revised version.
  122. Line 458-459 in the original version: “in the four conformations of hMTP and hMTP–E2 complexes after the MD simulations” is deleted in the revised version.
  123. Line 459 in the original version: “are generated” is replaced by “were generated” at line 453-454 in the revised version.
  124. Line 460-461 in the original version: “the four final conformations of hMTP and hMTP–E2 complexes in MD simulations.” is replaced by “the four conformations of hMTP and hMTP–E2 complexes after MD simulations. For convenience of description, SMD simulations starting from the conformations after MDs with and without a hydrogen bond constraint are named as the first and second SMDs, respectively.” at line 455-458 in the revised version.
  125. Line 466-474 in the original version: “Representative conformations during the steered molecular dynamics (SMD) simulation process. (A) Representative conformations of hMTP complex during SMD simulations which start from the conformations after MD simulations with a hydrogen bond constraint. (B) Representative conformations of hMTP–E2 complex during SMD simulations which start from the conformations after MD simulations with a hydrogen bond constraint. (C) Representative conformations of hMTP complex during SMD simulations which start from the conformations after MD simulations without the hydrogen bond constraint. (D) Representative conformations of hMTP–E2 complex during SMD simulations which start from the conformations after MD simulations without a hydrogen bond constraint.” is replaced by “Representative conformations of hMTP and hMTP–E2 complexes during the steered molecular dynamics (SMD) simulation process. For convenience, SMD simulations using the conformations after MDs with and without a hydrogen bond constraint are named as the first and second SMDs, respectively. (A) Representative snapshots of hMTP complex during the first SMDs. (B) Representative snapshots of hMTP–E2 complex during the first SMDs. (C) Representative snapshots of hMTP complex during the second SMDs. (D) Representative snapshots of hMTP–E2 complex during the second SMDs.” at line 463-469 in the revised version.
  126. Line 477 in the original version: “are extracted” is replaced by “were extracted” at line 473 in the revised version.
  127. Line 485-486 in the original version: “the SMD simulation process which starts from the conformation without a hydrogen bond constraint” is replaced by “the second SMDs” at line 480 in the revised version.
  128. Line 487 in the original version: “are fixed” is replaced by “were fixed” at line 481 in the revised version.
  129. Line 488 in the original version: “is fixed” is replaced by “was fixed, ” at line 482 in the revised version.
  130. Line 493-494 in the original version: “changes of the SMD forces with variations of the distance” is replaced by “changes of the distance with variations of the SMD forces” at line 488-489 in the revised version.
  131. Line 497-503 in the original version: “No distinct difference is observed for the SMD simulation results for the hMTP complex either in the presence or absence of E2 when the complex starts from a conformation (de-rived from the MD simulations) with a hydrogen bond constraint (Figure 6A). In comparison, distinct differences are observed for the hMTP and hMTP–E2 complexes which start from the respective conformations without hydrogen bond constraint (Figure 6B). When the same forces imposed on hMTPα subunit are reached,” is replaced by “No distinct difference is observed for the first SMDs results for the hMTP complex either in the presence or absence of E2 (Figure 6A). In comparison, distinct differences are observed for the hMTP and hMTP–E2 complexes during the second SMDs processes (Figure 6B). With the same SMD forces for the hMTP and hMTP–E2 complexes,” at line 492-496 in the revised version.
  132. Line 509-515 in the original version: “(A) Changes of distances between hMTPα subunit and hPDI with the SMD forces during the SMD simulation process for hMTP and hMTP–E2 The SMD simulations start from the con-formation after the MD simulations with a hydrogen bond constraint. (B) Changes of distances between hMTPα subunit and hPDI with the SMD forces during the SMD simulation process for hMTP and hMTP–E2 complexes. The SMD simulations start from the conformation after the MD simulations without the hydrogen bond constraint.” is replaced by “For convenience, SMD simulations using the conformations after MDs with and without a hy-drogen bond constraint are called as the first and second SMDs, respectively. (A) Changes of dis-tances between hMTPα subunit and hPDI with the SMD forces during the first SMDs. (B) Chang-es of distances between hMTPα subunit and hPDI with the SMD forces during the second SMDs.” at line 502-506 in the revised version.
  133. Line 520 in the original version: “are calculated” is replaced by “were calculated” at line 511 in the revised version.
  134. Line 522-524 in the original version: “between the SMD simulations for hMTP and hMTP–E2 complexes starting from the conformations (after the MD simulations) with a hydrogen bond constraint” is replaced by “between the first SMDs for hMTP and hMTP–E2 complexes” at line 513-514 in the revised version.
  135. Line 525-527 in the original version: “are derived from the SMD simulations of the first 1 ns which start from the conformations of the hMTP and hMTP–E2 complexes (after the MD simulations) without the hydrogen bond constraint.” is replaced by “were derived from the first 1 ns of the second SMDs for the hMTP and hMTP–E2” at line 515-516 in the revised version.

Reviewer 2 Report

In this manuscript, authors have analyzed nicely molecular modelling to show the hypothetical ER-independent lipid-modulating effect of E2 via the conformational changes in the hMTP complex. However, these results need to be validated by biochemical analysis to show that E2 is destabilizing PDI binding to MTP.

Author Response

RESPONSE: The present manuscript contains data for the molecular modeling analysis of the interaction of 17β-estradiol (E2) with hMTP. The modeling results offer a plausible mechanistic explanation for the estrogen receptor-independent lipid-modulating effect of E2. Guided by the initial results of this computational modeling study, we are presently conducting a series of experiments to analyze the ability of E2 as well as other compounds to bind to hPDI protein and to interfere with the interaction between hPDI and hMTP. So far we have obtained some experimental results (but incomplete at present) which are consistent with the computational modeling analysis presented in our present manuscript. We plan to publish these experimental data in the future as a separate, complete experimental study when a lot more data are collected. Summarized below are some of the experiments (along with representative results) that we have been doing lately:

       1) Purification of hPDI and hMTP: We have constructed two plasmids: one is pcDNA3.1-hMTP-His and the other one is pcDNA3.1-hPDI-Flag, and they were transfected into HEK-293T cells for expression. The selective expression and purification of hPDI has been completed, while the purification of hMTP is still ongoing.

      2) We have determined the ability of E2 to inhibit the reductase activity of purified hPDI protein in vitro using insulin as a substrate (see figure below). The inhibition of hPDI activity reflects the ability of E2 to bind to this protein and then inhibit its function. When the purified hMTP protein is ready, we plan to experimentally determine the hPDI-hMTP binding interactions in vitro, in particular the ability of E2 to alter their binding interactions in vitro.

   3) We are determining the interaction between hPDI and hMTP as well as their influence by E2 in intact live cells by using the pull-down assay. The constructed plasmids (containing hPDI and hMTP) were transfected into HEK-293T cells for expression. One hour before transfection, the cells were treated with 5 µM E2. Cells were collected 24 hours after transfection and Ni-NTA sepharose was used to pull down the hMTP protein from the cell lysates. The preliminary result from one of the experiments is shown below. This preliminary data clearly indicates that the presence of E2 can affect the interaction between hPDI and hMTP in intact live cells. We are validating this observation by performing more experiments under different treatment conditions (such as using different E2 concentrations).

Reviewer 3 Report

In this manuscript, entitled: “17β-Estradiol-Induced Conformational Changes of Human 2 Microsomal Triglyceride Transfer Protein: A Computational Molecular Modelling Study” (by Yong Xiao Yang, Peng Li, Pan Wang and Bao Ting Zhu) submitted to Cells for publication, the authors, based on known experimental structures, use molecular dynamics simulations (classic and steered MDs) to investigate the role of a female sex hormone (17β-estradiol, E2) in the stability of the human microsomal triglyceride transfer protein (hMTP), which is involved in the assembly of apoB-containing lipoproteins. By comparing several metrics calculated through various analyses of the simulations, the authors conclude that the presence of the E2 hormone affects the stability of the hMTP heterodimer by increasing the flexibility of the interface between its α and β subunits and propose this as a novel mechanism underlying lipid-modulating actions of E2 that are independent of estrogen receptors.

Overall this is an interesting study and contributes to a better understanding of the hMTP structure as drug target. However, I have a major comment on the manuscript:

The results obtained from analyses of the classical MD simulations and presented in Tables 1, 2 and S1, are based on snapshots at the end of the corresponding MD trajectories (at 50 ns). The same 50 ns snapshots are used as starting conformations for the corresponding SMD simulations. To my opinion, for more reliable results, it would be more appropriate to produce more than one MD replicas (50 ns each) for each simulated complex, especially due to the observed conformational flexibility of the C-terminal regions that however, contribute to the formation of the heterodimer interface.  Alternatively, at least, the metrics presented in Tables 1, 2 and S1, should be calculated from an ensemble of conformations, e.g. from the last 5 or 10 ns of the corresponding 50 ns MD simulations, rather than from the single 50 ns snapshot.

Minor points:

2.5.1 Which software is used for the calculation of the MSFD metric?

Lines 493-495: The words “SMD forces” and “distance” should be interchanged in the text to correspond to the plots shown in Fig 6.

Lines 502-503: “When the same forces imposed on hMTPα subunit are reached”, change phrasing.

In general, there are redundant word repetitions throughout the manuscript, especially in the Figure legends. For example, “after MD simulations” in the legend of Fig 2 or “Representative conformations during SMD simulations which start from the conformations after MD simulations” in the legend of Fig 5, should appear only once in the title, for clarity. In general, such word repetitions should be avoided throughout the manuscript and emphasis should be given on the different parts of such sentences, to facilitate reading of the manuscript

Syntax

  • 2.5, line 191: There is no verb in the first sentence. Is this a title?

Line 322: are shown

Lines 350-351: an “and” should be added:  “.. its interactions with hPDI and  decreases...”

Line 308: an “and” should be added:  “.. bond constraint and …”

Author Response

Overall this is an interesting study and contributes to a better understanding of the hMTP structure as drug target. However, I have a major comment on the manuscript:

       The results obtained from analyses of the classical MD simulations and presented in Tables 1, 2 and S1, are based on snapshots at the end of the corresponding MD trajectories (at 50 ns). The same 50 ns snapshots are used as starting conformations for the corresponding SMD simulations. To my opinion, for more reliable results, it would be more appropriate to produce more than one MD replicas (50 ns each) for each simulated complex, especially due to the observed conformational flexibility of the C-terminal regions that however, contribute to the formation of the heterodimer interface. Alternatively, at least, the metrics presented in Tables 1, 2 and S1, should be calculated from an ensemble of conformations, e.g. from the last 5 or 10 ns of the corresponding 50 ns MD simulations, rather than from the single 50 ns snapshot.

RESPONSE: We appreciate the reviewer’s comment and fully understand the underlying rationale. In this study, we used a slightly different computational paradigm, for the following reasons:

       1) It is commonly-accepted that to derive the potential of mean force (PMF) by using SMD simulations, the initial conditions need to be the same for at least 10 SMD simulations. Because of this consideration, in this study we initially did not choose to do the SMD simulations by using different initial conformations.

       2) Also considering the limited computational power and resources that were available to us, we opted to use the following modelling approach: We first generated four different MD trajectories, and then for each of these four MD trajectories we generated 10 SMD trajectories with the same initial conditions. This joint MD and SMD approach helped to simulate the dynamic behaviors of hMTP and hMTP-E2 complexes. We feel that the computational data produced in this study are adequate to help explain the mechanism for the experimental observations reported earlier.

       (Here it is of note that in another related work which is presently underway in our laboratory, we applied longer time of MD simulations to compare the effect of other compounds on the conformational changes of hPDI and hMTP complexes. In addition, multiple initial conformations of hPDI are also adopted in the MD simulations.)

       Regarding the metrics presented in Tables 1, 2 and S1, they were intended to be used as a simple reference, and we agree with the reviewer that they are not overly rigorous for reflecting and approximating the binding strength in protein-protein complex. However, in the ensuing part of the manuscript, a more intuitive and rigorous metric, MSFD, was also calculated, which would better reflect the stability of interface contacts in hMTP and hMTP-E2 complexes during the MD simulation processes. Jointly, these metrics provide a somewhat more complete picture of the static and dynamic behaviors of hMTP and hMTP-E2 complexes.

Minor points:

2.5.1 Which software is used for the calculation of the MSFD metric?

RESPONSE: The conformations are extracted from the MD trajectory using VMD [1]. A simple script is written to calculate the MSFD based on the conformations generated by using MATLAB.

References:

  1. Humphrey W, Dalke A, Schulten K. VMD: Visual molecular dynamics. Journal of Molecular Graphics & Modelling 1996, 14: 33-38

Lines 493-495: The words “SMD forces” and “distance” should be interchanged in the text to correspond to the plots shown in Fig 6.

RESPONSE: Done as suggested.

Lines 502-503: “When the same forces imposed on hMTPα subunit are reached”, change phrasing.

RESPONSE: “When the same forces imposed on hMTPα subunit are reached” has been replaced by “With the same SMD forces for the hMTP and hMTP–E2 complexes”.

In general, there are redundant word repetitions throughout the manuscript, especially in the Figure legends. For example, “after MD simulations” in the legend of Fig 2 or “Representative conformations during SMD simulations which start from the conformations after MD simulations” in the legend of Fig 5, should appear only once in the title, for clarity. In general, such word repetitions should be avoided throughout the manuscript and emphasis should be given on the different parts of such sentences, to facilitate reading of the manuscript

RESPONSE: The legends of Figure 2, 5 and 6 have been changed to remove redundant words. The manuscript has been checked and changed to eliminate other redundant words. Please see the detail described in the cover letter.

Syntax

2.5, line 191: There is no verb in the first sentence. Is this a title?

RESPONSE: “Number of Interface Atom Pairs and Areas of Interface Regions” is a sub-title, which has been changed according to the format.

Line 322: are shown

RESPONSE: The words “are show” have been replaced with “are shown”.

Lines 350-351: an “and” should be added:  “... its interactions with hPDI and  decreases ...”

RESPONSE: Revised as suggested.

Line 308: an “and” should be added:  “... bond constraint and …”

RESPONSE: Revised as suggested.

Round 2

Reviewer 2 Report

Authors have successfully responded to the queries raised.